# Differential Distribution and Activity Profile of Acylpeptide Hydrolase in the Rat Seminiferous Epithelium

**DOI:** 10.3390/biomedicines10071591

**Published:** 2022-07-04

**Authors:** Alejandra A. Covarrubias, Erwin De la Fuente-Ortega, Gabriela Rossi, Ennio Cocca, Mosè Rossi, Gianna Palmieri, Floria C. Pancetti

**Affiliations:** 1Laboratorio de Neurotoxicología Ambiental, Departamento de Ciencias Biomédicas, Facultad de Medicina, Universidad Católica del Norte, Coquimbo 1781421, Chile; alejandra.covarrubias@ucn.cl (A.A.C.); gabriela.rossi.v@gmail.com (G.R.); 2Departamento de Ciencias Biomédicas, Facultad de Medicina, Universidad Católica del Norte, Coquimbo 1781421, Chile; edelafuente@ucn.cl; 3Institute of Biosciences and Bioresources, National Research Council (CNR-IBBR), 80131 Naples, Italy; ennio.cocca@ibbr.cnr.it (E.C.); m.rossi@ibp.cnr.it (M.R.); gianna.palmieri@ibbr.cnr.it (G.P.); 4Centro de Investigación y Desarrollo Tecnológico en Algas y Otros Recursos Biológicos (CIDTA), Universidad Católica del Norte, Coquimbo 1781421, Chile

**Keywords:** spermatogenesis, acylpeptide hydrolase, spermatozoa, acrosome

## Abstract

Acylpeptide hydrolase (APEH) is a serine protease involved in amino acid recycling from acylated peptides (exopeptidase activity) and degradation of oxidized proteins (endoproteinase activity). This enzyme is inhibited by dichlorvos (DDVP), an organophosphate compound used as an insecticide. The role of APEH in spermatogenesis has not been established; therefore, the aim of this study was to characterize the distribution and activity profile of APEH during this process. For this purpose, cryosections of male reproductive tissues (testis and epididymis) and isolated cells (Sertoli cells, germ cells, and spermatozoa) were obtained from adult rats in order to analyze the intracellular localization of APEH by indirect immunofluorescence. In addition, the catalytic activity profiles of APEH in the different male reproductive tissues and isolated cells were quantified. Our results show that APEH is homogeneously distributed in Sertoli cells and early germ cells (spermatocytes and round spermatids), but this pattern changes during spermiogenesis. Specifically, in elongated spermatids and spermatozoa, APEH was localized in the acrosome and the principal piece. The exopeptidase activity was higher in the germ cell pool, compared to sperm and Sertoli cells, while the endoproteinase activity in epididymal homogenates was higher compared to testis homogenates at 24 h of incubation. In isolated cells, this activity was increased in Sertoli and germ cell pools, compared to spermatozoa. Taken together, these results indicate that APEH is differentially distributed in the testicular epithelium and undergoes re-localization during spermiogenesis. A possible role of APEH as a component of a protection system against oxidative stress and during sperm capacitation is discussed.

## 1. Introduction

Spermatogenesis is a finely coordinated process of cell proliferation and differentiation, with the ultimate goal of producing haploid motile cells. To generate mature spermatozoa, germ cells undergo mitosis (self-renewal and proliferation of spermatogonia), meiosis (generation of round spermatids), and spermiogenesis (transformation of round spermatids into sperm) [1,2,3]. Spermatogenesis involves several important events, including differentiation of primordial germ cells [4,5], signaling between germ and Sertoli cells [6,7], and substitution of histones by protamines after meiosis [8,9,10]. These physiological processes can be affected by a large number of exogenous agents that include heavy metals, phthalates, bisphenol-A, polychlorinated biphenyls, dioxins and pesticides, among other xenobiotics [11,12,13]. In many developed countries, there is increasing concern regarding the adverse effects of pesticide exposure on human reproduction [14,15,16]. These substances elicit different effects on the male reproductive system, such as testicular degeneration and low sperm concentration, which causes infertility [17,18]. Despite the evidence reported in the literature, the molecular targets involved, and the affected cellular mechanisms have not yet been identified.

A wide variety of serine proteases have been described in the seminiferous epithelium [19]. Serine proteases are involved in germ cell migration through the seminiferous epithelium, meiosis initiation, sperm maturation, capacitation, and fertilization [20,21,22,23,24,25]. Specifically, mRNA of prolyl-oligopeptidase (POP; EC 3.4.21.26) [24], a member of the POP superfamily of serine proteases, is expressed in mouse testis during postnatal development, and the enzyme activity remains in adult mice, restricted to spermatids at stages I–VIII of the spermatogenic cycle [26]. The same enzyme has also been detected in somatic cells, such as Sertoli cells [27]. On the other hand, the role of another POP family member, acylpeptide hydrolase (APEH), has not been investigated in this epithelium although this enzyme can be specifically and efficiently inhibited by dichlorvos (DDVP), an organophosphate pesticide [28,29]. APEH is an exopeptidase that catalyzes the hydrolysis of several peptides with an acylated *N*-terminal amino acid, yielding an acylated amino acid and a peptide with an acyl-free *N*-terminus [30,31]. In addition, APEH also displays an endoproteinase activity against oxidized proteins [32,33] which, in coordination with the proteasome, has been attributed to a role as a secondary antioxidant defense system through the removal of oxidized proteins [34,35]. Of note, both catalytic activities of APEH are inhibited by DDVP [36].

Several signs of sperm toxicity have been observed in the seminiferous epithelium of DDVP-treated rats, such as alterations in sperm morphology, cytoplasmic droplets, and reduced motility [18]. However, the molecular target affected by DDVP has not been identified in this model. 

In the present work, we sought to characterize the tissue distribution, intracellular localization, and activity profiles of APEH in the seminiferous epithelium, using cryosections of adult male rat reproductive tissues (testis and epididymis) and isolated cells (Sertoli cells, pool germ cells, and spermatozoa). Interestingly, we observed an evident change in both APEH protein distribution and enzyme activity patterns during spermiogenesis, suggesting antioxidative and sperm capacitation roles at specific stages of the spermatogenic cycle.

## 2. Materials and Methods

### 2.1. Animals

Tissue samples were obtained from 3- to 5-month-old Sprague Dawley rats. All rats were housed under a 12:12 h light–dark cycle with *ad libitum* food and water. Animals were anesthetized with isofluorane and sacrificed by decapitation. All protocols and animal handling techniques were performed in strict accordance with NIH guidelines and approved by the Ethics Committee of the Faculty of Medicine, Universidad Católica del Norte, according to Resolution #14 from 22 May 2017.

### 2.2. Primary Culture of Sertoli Cells

Primary cultures of Sertoli cells were obtained as described [37]. Briefly, testes were removed and decapsulated; seminiferous tubules were dispersed, but not fragmented, in 25 mL of 0.5 mg/mL collagenase (Sigma, St. Louis, MO, USA, C2674) in Hank’s Balanced Salt Solution (HBSS; Gibco, Waltham, MA, USA, 14025-092), pH 7.4, at 34 °C for 10–15 min under constant shaking (80 oscillations/min), after which the suspension was allowed to settle. Tubules were washed three times in Hank’s Balanced Salt Solution (HBSS) and then incubated in 25 mL of 0.5 mg/mL trypsin (T5266; Sigma) in HBSS, pH 7.4, for 5–10 min at 37 °C, without shaking. Tubules were then washed twice in HBSS, followed by the last wash in DMEM/F12 (Gibco, Waltham, MA, USA, 11320-033) supplemented with 10% fetal bovine serum (FBS; Biological Industries, Kibbutz Beit-Haemek, Israel, 04-127-1A) and tubules were allowed to settle. To separate Sertoli from germ cells, tubules were incubated in a solution containing 0.1% collagenase, 0.2% hyaluronidase (Sigma, St. Louis, MO, USA, H6254), and 0.04% DNase I (Sigma, St. Louis, MO, USA, D5024) in HBSS/1% FBS, pH 7.4, for 40 min at 34 °C with constant shaking (80 oscillations/min). The suspension was then centrifuged at 125× *g* for 2 min and the pellet was washed twice in HBSS and, finally, once in DMEM/F12/10% FBS. The germ cells adhered to Sertoli cells were mechanically removed by repetitive pipetting with a micropipette. The germ cell and Sertoli cell pools were used to measure APEH activity and for immunofluorescence experiments.

### 2.3. Cryosection Immunofluorescence

Epididymis and testis tissues were fixed in PBS/4% paraformaldehyde for 2 h and were then incubated in PBS supplemented with 30% sucrose for 48 h, mounted in optimal cutting temperature (OCT) embedding compound, and frozen at −20 °C to −80 °C. Tissue sections between 5 and 15 µm-thick were cut on a Microm HM525 cryostat (Thermo Scientific, Waltham, MA, USA) and cryosections were mounted on SuperFrost Plus microscope slides (Fisherbrand, Waltham, MA, USA, USA). Slides were washed in PBS and sections were encircled with a PAP pen. Tissues were then blocked for 1 h at room temperature (RT) in 10% horse serum in PBS plus 0.05% Triton X-100. The blocking solution was eliminated, and the slides were incubated overnight at 4 °C with anti-APEH primary antibody (1:200 rabbit; LS Bios LS-B5812), anti-serine/threonine sperm kinase 2 (TSSK2) (1:200 mouse; Santa Cruz Biotechnology, Santa Cruz, CA, USA, sc-100437) and anti-acrosin (1:200 mouse; Santa Cruz Biotechnology sc-51504), diluted in blocking buffer (PBS supplemented with 10% (*w/v*) BSA). Slides were washed three times in Dulbecco’s PBS (DPBS) supplemented with 0.9 mM CaCl_2_ and 0.5 mM MgCl_2_ (DPBS Ca/Mg) for 5 min. Secondary antibodies (anti-rabbit IgG-Alexa 488 [1:500, Molecular Probes A11008] and anti-mouse IgG-Alexa 568 [1:500, Molecular Probes A11004]) were incubated for 2 h in DPBS supplemented with 10% (*w/v*) BSA. Nuclei were stained with Hoechst 33342 (1:10,000; Invitrogen H1399).

### 2.4. Immunocytochemistry

Spermatozoa, Sertoli cells, and pooled germ cells were stained with MITOTRACKER RED (CMXRos; Cell Signaling Technology #9082) for 15 min at 37 °C and then fixed for 10 min in PBS/4% PFA. Cells were then washed 3 times for 5 min in DPBS Ca/Mg and permeabilized for 10 min in PBS/0.02% Triton X-100. After incubation in blocking buffer, primary antibodies were incubated overnight in blocking buffer, and cells were then washed 3 times for 5 min in DPBS Ca/Mg. Secondary antibodies were incubated for 2 h in blocking buffer. After washing in PBS, cells were mounted onto glass slides in ProLong fluorescence mounting medium (Invitrogen P36965). Anti-APEH (1:200), anti-SCP3 (1:200 mouse; Santa Cruz Biotechnology sc-74569), anti-TSSK2 (1:200), and anti-GM130 (1:100 mouse; BD Biosciences 610822) were used as primary antibodies and anti-rabbit IgG-Alexa-Fluor-488 and anti-mouse IgG-Alexa-Fluor-568 were used as secondary antibodies. Nuclei were stained with Hoechst 33342 (1:10,000). 

### 2.5. Fluorescence Imaging

Fluorescence images were taken using a Zeiss LSM 800 confocal microscope (Carl Zeiss, Heidelberg, Germany) under a Plan-Apochromat 63 ×/1.46 oil objective. Images were acquired as 16-bit, avoiding signal saturation, pinhole adjusted to 1 Airy unit, gain between 630 and 770 V, and laser power ranging from 1.48 to 21.32% for the 488 nm laser. Acquired images were processed generating regions of interest (ROIs) using ZEN Imaging Software 3.4 (Carl Zeiss, Heidelberg, Germany), and figure composition was performed using Adobe Photoshop CS6 (Adobe Systems, San José, CA, USA). Signal colocalization was quantified using the Manders’ coefficients method as described previously [38]. To establish the area occupied by APEH (Alexa Fluor 488), we measured the pixel number of markers (APEH) that colocalized with acrosin signal and divided these values by the total pixels obtained for each marker.

### 2.6. Homogenization of Tissues and Cells and Protein Quantification

For immunoblot analysis, homogenates were obtained from tissue samples (epididymis, testis, brain, liver, and kidney) and isolated cells (Sertoli cells, germ cells, and spermatozoa), using a lysis buffer containing 0.05 M Tris-HCl, pH 7.4, 0.1 M NaCl, 0.05 M MgCl_2_, 1% Triton X-100, 5 µg/mL each of aprotinin, leupeptin, soybean trypsin inhibitor and 1 mM phenylmethylsulfonyl fluoride (PMSF). For exopeptidase activity assays, tissue and cell homogenates were obtained using a cold buffer containing 50 mM Tris-HCl, pH 7.4, 1 M NaCl, 50 mM MgCl_2_, and 1% Triton X-100. For endoproteinase activity assays, homogenates were obtained by incubating the samples in ice-cold extraction buffer (10 mM Tris-HCl, pH 7.5, 150 mM NaCl, 0.5% Nonidet P40 and 1 mM PMSF). In all three procedures, homogenization of tissues was accomplished by passing the samples through a 21-gauge syringe needle. Each lysate was obtained from at least three different animals. Finally, total protein concentration was determined using the bicinchoninic acid (BCA) assay [39]. 

### 2.7. Immunoblotting

Protein samples (30 µg/lane) were run on 10% SDS-polyacrylamide gels and proteins were then transferred to a polyvinylidene fluoride (PVDF) membrane (PerkinElmer) using a Mini Trans-Blot® (Bio-Rad) device. Blocking was performed in TBS-Tween-20 (0.01%) supplemented with 5% skim milk for 30 min and then membranes were incubated with anti-APEH (1:1000), anti-β−actin (Mouse, 1:10,000, Sigma Aldrich) or anti-α-tubulin (1:1000, Santa Cruz Biotechnology) overnight at 4 °C. Membranes were washed and incubated for 2 h at 37 °C with secondary antibodies conjugated to horseradish peroxidase (anti-rabbit IgG, 1:5000 GE Healthcare; anti-mouse IgG, 1:5000, Rockland). Specific bands were visualized by SuperSignal West Pico Plus Chemiluminescent Substrate (Thermo Scientific 34580). Quantitative immunoblotting analysis was carried out using Image J software (National Institutes of Health, Bethesda, MD, USA). 

### 2.8. Exopeptidase Activity

Measurement of exopeptidase APEH activity was performed by monitoring the hydrolysis of the synthetic substrate N-acetyl-L-alanine p-nitroanilide (AANA), as described previously [40]. Briefly, samples (10 µL containing 100–200 µg of protein) were mixed with 50 mM phosphate buffer at pH 8.0 in a total volume of 1 mL and incubated at 37 °C in a water-jacketed cuvette holder. The reaction was initiated by adding 10 μL substrate (1 nmol) dissolved in dimethyl sulfoxide. The final volume of the reaction mix was 1020 µL. Released *p*-nitroaniline was quantitated in a Specord 205 spectrophotometer (Analytik Jena, Germany) by measuring the absorbance increase at 410 nm using ε_410_ = 8800 M^−1^ cm^−1^. Enzymatic activity was normalized to protein content in each assay. For measurement of APEH activity using the fluorescent substrate acetyl-methionine-7 amino-4-methylcoumarin (Ac-Met-AMC) [41], 90 μL of 50 mM Tris-HCl pH 7.4 plus 1% triton X-100 were added to a 96 well plate. Then, 5 μL of the sample was added and incubated for 10 min at 37 °C. Measurement of the activity started when 5 μL of 10 mM Ac-Met-AMC were added to each well on the plate. Measurements of fluorescent emission were performed at 470 nm every 30 s for 5 min. For quantification, we assumed a molar extinction coefficient of 1 and we applied the following formula:(1)Activity(U/mL)= ( ΔA/t×Vt)/(1×Vs)where *Vt* = Total volume of the reaction mixture and *Vs* = Sample volume.

### 2.9. Obtention of Semi-Purified APEH Fractions

The “soluble fraction” was obtained by centrifugation of the lysate at 9000× *g* for 1 h at 4 °C. APEH was semi-purified from the supernatant following the method described by Gogliettino et al. (2014) [41], which uses two different resins. Briefly, the supernatant was loaded onto an Econo-column (Biorad 737-2532) with DEAE-Sephadex A-25 resin (Sigma Aldrich #A25120-50G), previously equilibrated in 25 mM Tris/HCl (pH 7.4) (buffer A). Proteins were eluted with an ionic strength gradient from 0 to 1 M NaCl in buffer A at a flow rate of 1mL min^−1^. The APEH active fractions were detected by measuring exopeptidase activity with the specific fluorogenic substrate Acetyl-Methionine-7-amino-4-methylcoumarin (Ac-Met-AMC). These fractions were pooled and dialyzed against buffer A at 4 °C for 48 h. Then, the total dialyzed fractions were centrifuged and concentrated at 5000× *g* for 15 min in Amicon^®^ ultra-15 centrifugal filters (Millipore, 30K). These concentrated samples were loaded onto an Econo-column^®^ (Biorad 737-2532) with the second resin, Superdex G200 (Pharmacia Biotech), equilibrated in 25 mM Tris/HCl, pH 7.4. The column was connected to a peristaltic pump and eluted with 25 mM Tris/HCl, pH 7.5, and 50 mM NaCl at a flow rate of 0.1 mL min^−1^. The APEH active fractions were dialyzed and concentrated as described above. Finally, the semipurified APEH was stored in 25 mM Tris/HCl, pH 7.4, at −20 °C.

### 2.10. APEH Endoproteinase Activity Assay

A solution of unoxidized or oxidized BSA (0.5 mg/mL), prepared as described by Fujino et al. (1998) [32], was incubated with 600 total units, determined with the substrate Ac-Met-AMC, of semi-purified APEH in 25 mM Tris/HCl, pH 7.4. The mix was incubated at 37 °C for 0, 12, 24, and 48 h. Then, 30 µg- protein samples were subjected to SDS/PAGE analysis. Assays were performed in triplicate. 

### 2.11. Statistical Analysis

Data were analyzed using the PRISM software (GraphPad Software Inc., San Diego, CA, USA). Significant differences were analyzed using the non-parametric Mann–Whitney U test. For analysis of the bands obtained by the endoproteinase activity assay, 2-way ANOVA and Bonferroni post-test were used. A probability level of 0.05 or less was considered significant. For all cases, significance level was established at α = 0.05, and defining 0.01 ≤ *p* < 0.05 as statistically significant (*), 0.001 ≤ *p* < 0.01 as highly significant (**), and *p* < 0.001 as whole level significance (***).

## 3. Results

### 3.1. Localization of APEH in Testis and Epididymis

The expression of APEH protein in testis and epididymis was assessed by Western blot and immunofluorescence. First, we determined the expression levels of APEH protein by Western blot across several rat tissue types. Figure 1A shows differential expression of the 75 kDa APEH protein band in testis, epididymis, brain, liver, and kidney, revealing the highest expression level in the latter two tissues (*p* < 0.05). However, in a saline-perfused rat, kidney APEH levels dropped sharply, while remaining constant in other tissues (see Appendix A), suggesting that in the kidney, a significant amount of APEH could come from the blood contained in vessels. 

The histological localization of APEH was studied in testis and epididymis cryosections by immunofluorescence. APEH signal was detected in all cell types, but it was stronger in the sperm tail in both tissues (Figure 2C–H). As observed in Figure 2B, the APEH signal showed a change in its distribution pattern, appearing as aggregates in some areas of the seminiferous tubules, depending on the spermatogenic stage. In order to distinguish between spermatogenic stages, we used antibodies against TSSK2 (testis-specific serine/threonine-protein kinase 2) which accumulates in a punctated pattern in elongated spermatids [42,43,44], allowing us to discriminate between stages I–VIII and IX–XIV. In testis cross-sections, the tubules with only elongated spermatids (Stages IX–XIV, Figure 2A,B), were positive for TSSK2 (red dots, Figure 2B), whereas APEH (green signal) was observed in a punctate pattern close to the lumen. On the other hand, in tubules with no TSSK2 signal, the APEH-labeled cells probably correspond to round and final-stage spermatids (stages I–VIII; Figure 2C,D). These spermatids did not display the punctate APEH signal detected in Figure 2A,B, but rather an elongated signal corresponding to spermatid tails (Figure 2C,D). Furthermore, in these tissues, we also used anti-acrosin, which labels the acrosomal vesicle (Figure 2F). In the testicle, both the APEH signal from tail structures and the acrosine signal appeared in the same spermatogenic stages (Figure 2E,F). This result was corroborated by spermatozoa observed in the epididymis where the same pattern was noticed (Figure 2G,H). To corroborate the specificity of the fluorescent signals, a negative control was included by using a blocking peptide against the APEH antibody, where the green signal was completely canceled in epididymis samples (Appendix A).

### 3.2. Localization of APEH in Germ Cells and Sertoli Cells

In order to study the localization of APEH in germ and Sertoli cells, we used a primary culture of Sertoli cells and a pool of detached germ cells. Spermatozoa extracted from the epididymal cauda were also investigated. APEH expression, as well as subcellular localization, was studied in these cells, using Western blot and immunofluorescence. We found that APEH is expressed in the pooled germ cells, Sertoli cells, and spermatozoa (Figure 3A), with higher protein levels in Sertoli cells, compared to spermatozoa (Figure 3B, *p* < 0.05).

Sertoli cells displayed a homogeneous APEH localization signal, as observed by immunofluorescence (Figure 4A). Part of this signal colocalized with the Golgi apparatus marker GM130 [45] (Figure 4A). Spermatocytes (Figure 4B, right) and round spermatids (Figure 4B, left) also showed a homogeneous APEH distribution. Spermatocytes were identified using the meiosis marker synaptonemal complex protein 3 (SCP3) [46]. In elongated spermatids, APEH appears localized around the nucleus (Figure 4C). TSSK2 signal in the testis displayed a punctate structure in this cellular type. APEH localization was similar in mature spermatozoa (Figure 4D). Unexpectedly, the APEH signal was observed only in the tail principal piece, which corresponds to axoneme structures and the fibrous sheath [1]. We did not detect colocalization with APEH in the midpiece when using the specific mitochondrial in vivo marker MITOTRACKER RED (Figure 4D). In addition to the principal piece, we also detected a high-intensity green APEH signal in the apical zone of spermatozoa heads (Figure 4D,E). The acrosomal localization of this signal was corroborated using anti-acrosin (Figure 5A). Colocalization of APEH with acrosin was determined using Mander’s colocalization coefficients (MCC^43^), which showed a coincidence of 59.4% ± 2.1%, while acrosin colocalized with APEH in 73.9% ± 2.2% (Figure 5B). 

### 3.3. Determination of APEH Exopeptidase and Endoproteinase Activities

The functionality of APEH was estimated by measuring the exopeptidase and endoproteinase activities in the testis, epididymis, germ, and Sertoli cells (Figure 6). In tissues, levels of exopeptidase activity did not show significant differences (*p* = 0.9; Figure 6A). To corroborate the specificity of the measured enzyme activity, we used DDVP as a specific APEH inhibitor. Appendix A shows that the presence of 1μM DDVP produced a strong decrease in the exopeptidase activity for all the tissue extracts analyzed, while extracts obtained from isolated germ cells displayed lower values of exopeptidase activity compared to tissues. We observed significantly higher exopeptidase activity in extracts of samples from the pool of germ cells compared to those from Sertoli cells (*p* < 0.05, one-way ANOVA) (Figure 6B).

A significant increase in endoproteinase activity was found in extracts of epididymis compared to testis after 24 h of incubation with oxidized BSA (*p* < 0.05; Figure 6C,D). In isolated cells (germ cells, spermatozoa, and Sertoli cells), endoproteinase activity was significantly increased in Sertoli and germ cells compared to spermatozoa (*p* < 0.001, Figure 6C,E). As above, the specificity of the catalytic reaction was corroborated using 1 μM DDVP as an APEH inhibitor, which, as expected, significantly decreased endoproteinase activity (*p* < 0.001) (Appendix A). 

## 4. Discussion

In this study, we report for the first time the presence of functional APEH in rat reproductive tissues (testis and epididymis) and isolated cells from the male reproductive tract (Sertoli cells, germ cells, and sperm). We observed that in somatic cells (Sertoli cells) and early germ cells such as spermatocytes, APEH is localized both in the nucleus and in the cytoplasm. This intracellular localization pattern was similar to that observed previously for POP (EC 3.4.21.26), another member of the POP family [27]. According to that study, the POP signal is found in both the nucleus and the cytoplasm of proliferative cells such as gonocytes and spermatogonia, and also in Leydig and Sertoli cells. The nuclear localization of POP has also been described in other proliferative tissues [47], suggesting a role for this enzyme in DNA synthesis. In particular, this was observed in the mouse brain during neurogenesis [48]. Taken together, these antecedents could indicate a possible role of APEH in some nuclear processes as suggested by a recent publication reporting APEH involvement in DNA repair through interaction with the scaffold protein XRCC1 [49].

Our results also indicate that APEH changes its intracellular distribution pattern during spermatogenesis. In early stage cells such as spermatocytes and round spermatids, the APEH immunodetection signal appears homogeneously distributed, while in spermatids it becomes concentrated in punctate signals around the nucleus. This becomes evident in mature spermatozoa, in which a strong signal is observed in the principal piece of the tail and in the acrosome. This redistribution of APEH during spermatogenesis is similar to that reported previously by Kimura et al. (2002) [24], who described the differential expression of POP mRNA throughout the mouse spermatogenic process. According to our results, there is no change in APEH protein expression levels when comparing different tissues and cells by Western blot. This finding suggests that the changes in the distribution pattern observed during spermatogenesis should be due to the re-localization of the enzyme. 

The APEH signal observed in the principal piece of the tail has not been previously described for the POP enzyme [24,27], in which the signal is distributed along the flagellum, with higher intensity in the midpiece, where it is associated with the cytoskeleton through tubulin interactions [27]. We discarded the presence of APEH in the midpiece since it does not co-localize with MITOTRACKER RED (Figure 4D,E), a specific marker for mitochondria, which are concentrated on this structure. The role of APEH in the principal piece remains to be clarified, although previous reports have linked APEH function to proteasome activity. Using a set of natural and synthetic APEH inhibitors, Palmieri et al. (2011) [34] showed that inhibition of APEH activity induces cell death without cytotoxicity by a mechanism involving downstream inhibition of the proteasome [34,35,50]. In sperm, the proteasome has been encountered in the flagellum, where the presence of several components of the ubiquitin-proteasome system (UPS) has been described [51,52,53]. Additionally, different UPS components such as psmc3 and Rnf19A have been observed in the inner and outer acrosomal membranes [51]. Furthermore, the accumulation of a variety of proteins in capacitated spermatozoa under proteasomal inhibiting conditions [54] reinforces the relevancy of proteasome-mediated events during fertilization. In fact, capacitated sperm require UPS components for the regulation of interactions with the zona pellucida (ZP), which in turn remodels the plasma membrane of the sperm and the acrosome before fertilization [54,55,56]. It is known that the UPS is able to process a variety of proteins, depending on the degree of sperm capacitation [52,55,56]. Thus, considering that the activity of APEH regulates the proteasomal activity and both proteins display a similar localization pattern in sperm, the possibility that APEH could be involved in the mechanisms underlying sperm capacitation and remodeling of the proteins on the acrosome membranes should not be discarded. In fact, we have detected the presence of APEH in immunoprecipitated supernatant of an in vitro assay in which the acrosomal reaction was induced on capacitated spermatozoa. This suggests that APEH is exocytosed during the acrosomal reaction and could play a role during fertilization [57].

A recent report [58] hypothesizes a role for APEH as a regulator of plasma membrane localization of a Ras GTPase, demonstrating that inhibition of this enzyme prevents functioning endosomal recycling, that is, of sorting and re-exporting of internalized constituents of the membrane. In the same direction, enhancement of long-term potentiation in rat hippocampal slices triggered by specific inhibition of APEH by an organophosphate [29] is linked mechanistically to a reduction of recycling endosome transport [59]. Hence, the involvement of APEH in the capacitation process, which involves reconstitution of the plasma membrane, appears plausible. In addition, the distribution of the APEH signal along with the principal piece, where the most important channel proteins are distributed, like Catsper [60], may be linked to major changes in the plasma membrane involving these factors during sperm capacitation.

An important issue is a possible relationship between cellular metabolism and APEH activity. Germ cells display a high proliferation rate in the early stages, which requires a large number of nucleotides available for genome replication and proteins and lipids for organelle duplication. In addition, the high proliferation rate leads to high oxygen consumption and the production of reactive oxygen species [61,62,63]. Specifically, spermatogonia use glucose as the primary source of energy and, starting from the spermatocyte differentiation stage, lactate consumption begins. Among all germ cells, spermatozoa exhibit the highest glycolytic activity and the lowest tricarboxylic acid (TCA) cycle activity, using only glucose or fructose for energy metabolism [64]. In summary, the seminiferous epithelium displays a high rate of cell division, which implies high rates of mitochondrial oxygen consumption by germ cells and poor vascularization, resulting in low oxygen tension in the testis [59] which, in turn, renders the seminiferous epithelium vulnerable to oxidative stress. This basal condition triggers a series of mechanisms that protect the tissue from free radical-mediated damage. Among them, APEH could be participating in aberrant protein quality control [59], thus protecting the cells from damage. In this sense, it is interesting to note that both the exopeptidase and endoproteinase APEH activities are higher in germ cells than in other cells and tissues, which could be indicating its relevance in amino acid recycling from peptides carrying an acylated amino acid at the N-terminus (exopeptidase activity) or from oxidized peptides or proteins (endoproteinase activity). In contrast to proliferating germ cells, Sertoli cells are highly differentiated and also exhibit levels of endoproteinase activity similar to germ cells. Although this observation seems contradictory, a possible explanation could be related to the metabolic characteristics of these cells, which obtain energy mainly from beta-oxidation of fatty acids, instead of aerobic glycolysis [65,66]. This high oxidative activity enhances oxidative stress, which in turn could explain the high level of endoproteinase activity observed. 

## 5. Conclusions

Although the precise physiological role of APEH remains to be elucidated, our findings could provide a basis for future studies about APEH function in the seminiferous epithelium and its relevance as a target to study the impact of some of the most widely-used environmental contaminants, such as pesticides and herbicides, to male fertility.

## Figures and Tables

**Figure 1 biomedicines-10-01591-f001:**
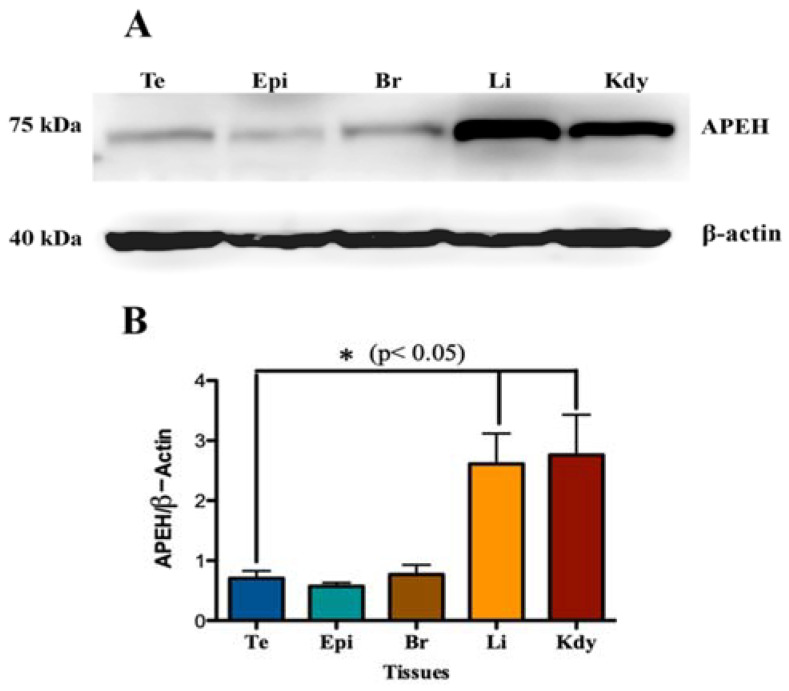
APEH levels in different rat tissues. (**A**) Protein extracts from testis (Te), epididymis (Epi), brain (Br), liver (Li), and kidney (Kdy) were obtained from male rats. Western blot analysis was performed with 30 µg of each extract, using an anti-APEH antibody. (**B**) Signal intensity of APEH was quantified with Image J and normalized against the signal intensity of β-actin. Each data point represents mean ± SEM of six experiments. Significant differences were found with non-parametric one-way ANOVA (* *p* < 0.05).

**Figure 2 biomedicines-10-01591-f002:**
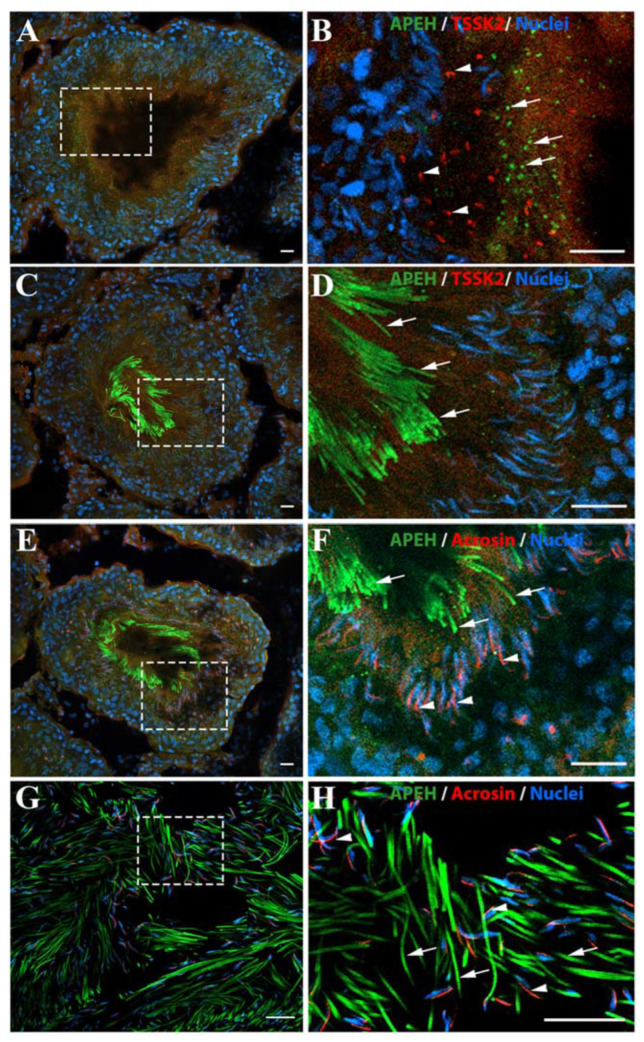
Distribution of APEH in testis and epididymis from male rats. Testis and epididymis cryosections were subjected to immunofluorescence against APEH (green). (**A**) Seminiferous tubule representative of stages IX–XIV, using an antibody against TSSK2 (testis specific serine/threonine-protein kinase 2, red) as an early spermatid marker. (**B**) Selected zoom of the immunofluorescence in A. White arrows show the APEH signal in punctate structures and white arrowheads indicate TSSK2 signal in punctate structures. No colocalization of TSSK2 with APEH is observed. (**C**) Seminiferous tubule representative of stages I-VIII using the TSSK2 marker. (**D**) Selected zoom of the immunofluorescence in C. White arrows show the APEH signal in the tails of elongated spermatids in final maturation stage. However, the TSSK2 signal was lost in these stages. (**E**) immunofluorescence in stage I-VIII seminiferous tubules using APEH (green) and acrosin (Red) antibodies. (**F**) Selected zoom of the immunofluorescence in E. Acrosin is localized to the heads of elongated spermatids in final maturation stage (white arrowheads), while, as before, APEH signal was observed in the tails (white arrows). (**G**) Epididymal cauda immunofluorescence against APEH and acrosin. (**H**) Selected zoom of immunofluorescence in G. APEH and acrosin signals observed in elongated spermatids were maintained in the epididymis. Nuclei were stained with Hoechst 33342 (blue). Scale bar = 20 µm.

**Figure 3 biomedicines-10-01591-f003:**
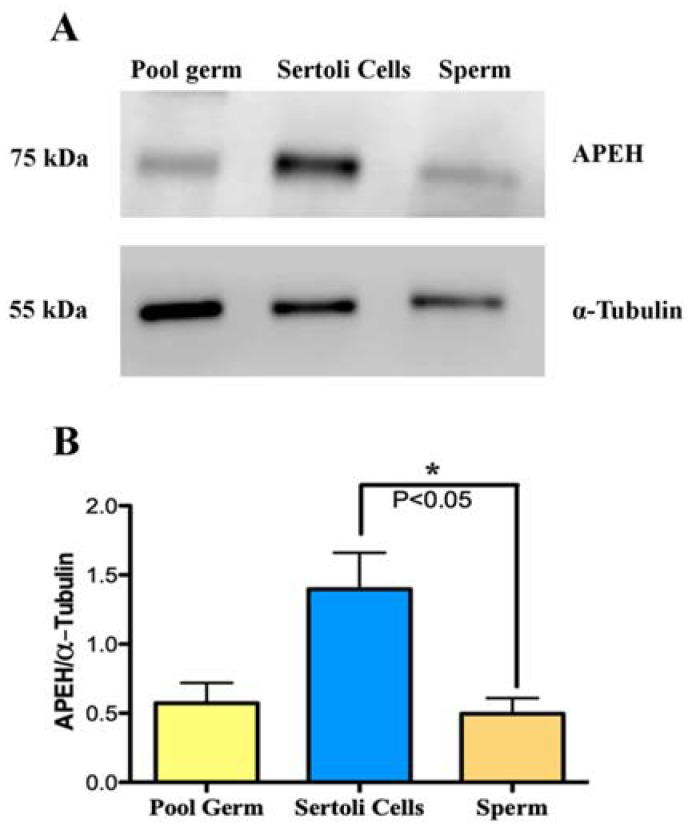
APEH protein levels in isolated cells obtained from male rat reproductive tissues. (**A**) Protein extracts from pooled germ cells, Sertoli cells, and sperm were obtained from primary cultures and epididymal cauda from male rats, respectively. Extracts (30 µg) were fractionated by SDS-PAGE, transferred to PDVF membranes, and subjected to Western blot using APEH antibody. (**B**) The signal intensity of APEH was quantified with Image J and normalized against the signal intensity of α-Tubulin. Data points represent mean ± SEM of three independent experiments. Significant differences were found with non-parametric one-way ANOVA (* *p* < 0.05).

**Figure 4 biomedicines-10-01591-f004:**
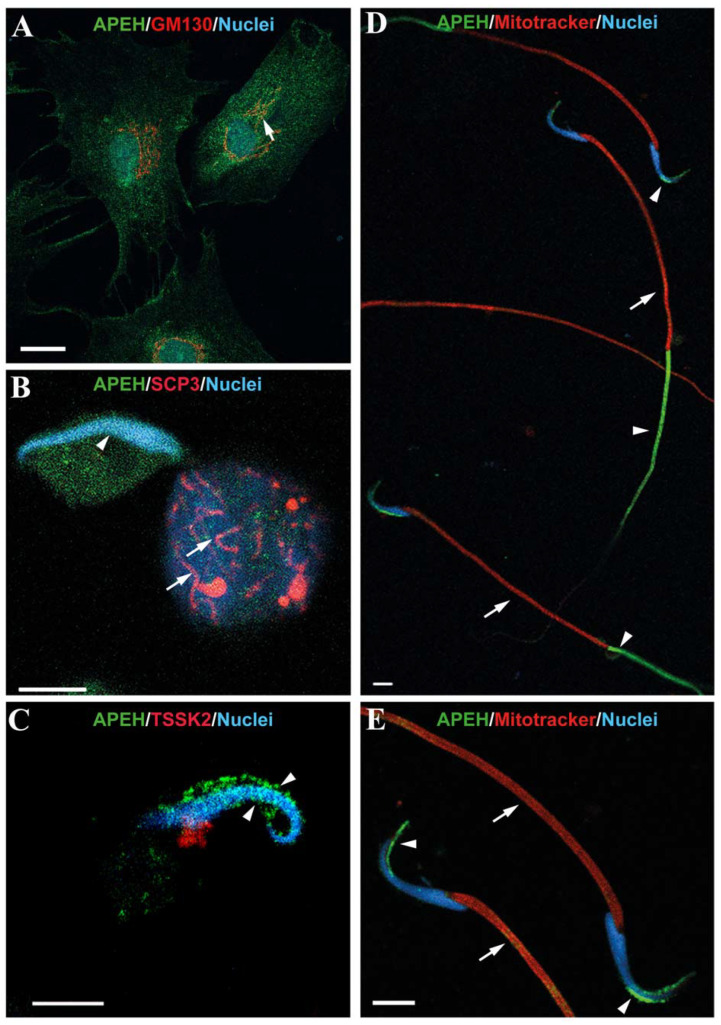
Subcellular distribution of APEH in isolated cells obtained from male rat reproductive tissues. Immunofluorescence against APEH (green) in Sertoli cells (**A**), germ cells (**B**), and spermatozoa (**C**) obtained from primary cultures and epididymal cauda from male rats, respectively. (**A**) a homogeneous cytoplasmic APEH signal is observed in Sertoli cells. The white arrow indicates partial colocalization of APEH with GM130 (*cis*-Golgi marker, red). (**B**) In early germ cells, a homogeneous distribution of APEH signal was observed. The white arrows indicate the distribution of SCP3 meiosis-specific marker (Synaptonemal complex protein 3, red) and white arrowheads show the nucleus of a round spermatid. (**C**) APEH signal in elongated spermatids. The white arrowheads show the differential distribution of APEH surrounding the nucleus of the elongated spermatid. (**D**) The APEH signal is distributed surrounding the upper part of the nucleus and in the principal piece of the sperm tail (white arrowheads). MITOTRACKER RED was used to identify the midpiece (white arrows). (**E**) Zoom of the sperm heads shown in 4D. APEH signal is distributed around the nucleus (white arrowheads) and MITOTRACKER RED in the midpiece (white arrows). Nuclei were stained with Hoechst (blue). Scale bar = 20 µm.

**Figure 5 biomedicines-10-01591-f005:**
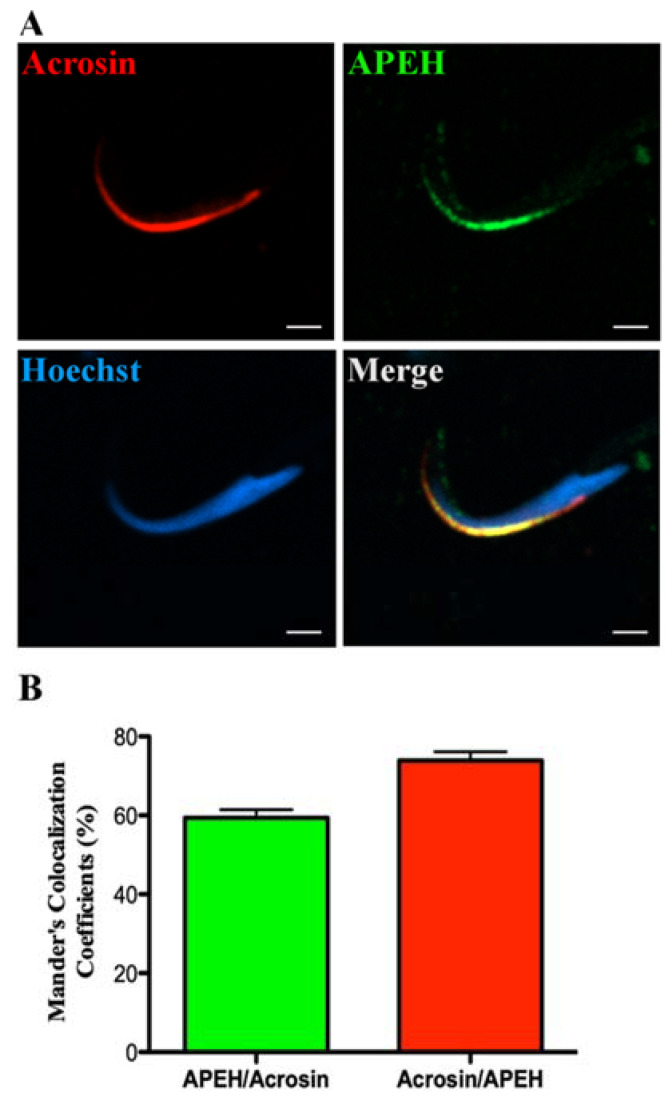
APEH colocalizes with Acrosin in rat sperm heads. (**A**) Immunofluorescence against APEH (green) and Acrosin (red) in spermatozoa obtained from epididymal cauda of male rats. (**B**) Quantification of APEH colocalization with Acrosin. Twenty-one sperm were checked for the determination of Mander’s colocalization coefficient. Nuclei were stained with Hoechst 33342 (blue). Scale bar = 2 µm.

**Figure 6 biomedicines-10-01591-f006:**
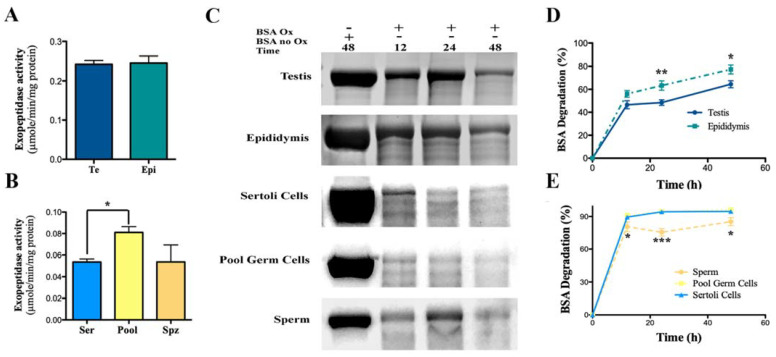
APEH activity in reproductive tissues and isolated cells. (**A**,**B**) Protein extracts from testis (Te), epididymis (Epi), Sertoli cells (Ser), pooled germ cells (Pool) and Spermatozoa (Spz) were obtained from male rats. Exopeptidase activity was measured at 37°C using N-acetyl-l-alanine p-nitroanilide (AANA) as substrate and expressed as specific activity. The results shown are representative of three independent experiments. (**C**) Endoproteinase activity was measured in tissues (Testis and Epididymis) and isolated cells (Sertoli, pooled germ, and spermatozoa) obtained from male rats. SDS-PAGE analysis of oxidized BSA treated at 37 °C with APEH purified from tissues and cells. (**D**,**E**) Electrophoretic data expressed as BSA degradation percentage at the indicated incubation times versus time 0, obtained by densitometric analysis with Image J software. The results shown are representative of three independent experiments. Significant differences were found with non-parametric one-way ANOVA (* *p* < 0.05; ** *p* < 0.01; *** *p* < 0.001).

## Data Availability

The data that support the findings of this study are available in the methods and/or Appendix A of this article.

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
