# Peer review of "Differential Distribution and Activity Profile of Acylpeptide Hydrolase in the Rat Seminiferous Epithelium"

_biomedicines, 2022, doi:10.3390/biomedicines10071591_

Round 1

Reviewer 1 Report

The publication is a typical descriptive work

Author Response

Thank you for your comment.  As authors we are aware that this is a descriptive report that shows for the first time the localization of APEH in the rat seminiferous tubule.  We believe that these set of results are useful to set a basis for further studies.

Reviewer 2 Report

The article “Differential distribution and activity profile of acylpeptide hydrolase in the rat seminiferous epithelium” reports acylpeptide hydrolase (APEH) distribution in male reproductive tissues and isolated cells. It shows that APEH locates in most of the reproductive cells at different stages implying it is involved in many activities of spermatogenesis and spermiogenesis. This paper would be of interest of estimation of DDVP usage and male infertility.

Author Response

Thank you very much for your comment. 

Reviewer 3 Report

The manuscript by Covarrubias and colleagues reports for the first time the presence of functional Acylpeptide Hydrolase (APEH) in rat reproductive tissues (testis and epididymis) and isolated cells from the male reproductive tract (Sertoli cells, germ cells and sperm). The authors also convincingly show that in Sertoli cells and spermatocytes APEH is localized both in the nucleus and in the cytoplasm and that the intracellular of the enzyme is changed during spermatogenesis.

The relevance of these findings in the “hypothetical” function of APEH is widely discussed, but no direct experimental evidence is provided.

Author Response

Thank you for your comment.  As authors we are aware that this is a descriptive report that shows for the first time the localization of APEH in the rat seminiferous tubule. Furthermore, in the last paragraph of the discussion we stated that the physiological role of APEH in the seminiferous tubule remains to be elucidated, but despite of this, we believe that these set of results are useful to set a basis for further studies.  We are preparing another manuscript in which we describe the presence of APEH in the supernatant of an in vitro assay in which acrosomal reaction was induced in capacitated spermatozoa, indicating that the enzyme is secreted during this process and therefore, highlighting the relevancy of its presence in the acrosome where it could be playing a role during fertilization.  We have added a phrase in the discussion explaining this and pointing out that the manuscript is in preparation.  

Reviewer 4 Report

This manuscript evaluates the role of APEH in spermatogenesis. In particular, the authors demonstrated that this serine protease is involved in the rat control of spermatogenesis and in particular spermiogenesis process. The authors have shown very good quality images in this manuscript, where the signals are clearly evident. This study is an interesting and potentially valuable addition to the literature on the control of spermatogenesis. Before the manuscript can be accepted for publication, a small revision is necessary to improve its quality.

Line 48: In the introduction, the authors in describing the effects of xenobiotics on spermatogenesis did not take into account two very recent papers, which they should cite in the text.

-        Verderame, M.; Chianese, T.; Rosati, L.; Scudiero, R. Molecular and Histological Effects of Glyphosate on Testicular Tissue of the Lizard Podarcis siculusInt. J. Mol. Sci. 202223, 4850. https://doi.org/10.3390/ijms23094850

-        - Di Lorenzo, M.; Mileo, A.; Laforgia, V.; De Falco, M.; Rosati, L. Alkyphenol Exposure Alters Steroidogenesis in Male Lizard Podarcis siculusAnimals 202111, 1003. https://doi.org/10.3390/ani11041003

In the testis immunofluorescence results, the authors show higher immunopositivity for APEH in spermatozoa compared to Sertoli cells. They then showed a blot in which a higher quantity of this protease is evident in the Sertoli than in the spermatozoa. How do the authors explain this result?

In Figure 5A, the authors should put the bars in all figures 

Author Response

Reviewer comment:

"This manuscript evaluates the role of APEH in spermatogenesis. In particular, the authors demonstrated that this serine protease is involved in the rat control of spermatogenesis and in particular spermiogenesis process. The authors have shown very good quality images in this manuscript, where the signals are clearly evident. This study is an interesting and potentially valuable addition to the literature on the control of spermatogenesis. Before the manuscript can be accepted for publication, a small revision is necessary to improve its quality.

Line 48: In the introduction, the authors in describing the effects of xenobiotics on spermatogenesis did not take into account two very recent papers, which they should cite in the text.

-        Verderame, M.; Chianese, T.; Rosati, L.; Scudiero, R. Molecular and Histological Effects of Glyphosate on Testicular Tissue of the Lizard Podarcis siculusInt. J. Mol. Sci. 202223, 4850. https://doi.org/10.3390/ijms23094850

-        - Di Lorenzo, M.; Mileo, A.; Laforgia, V.; De Falco, M.; Rosati, L. Alkyphenol Exposure Alters Steroidogenesis in Male Lizard Podarcis siculusAnimals 202111, 1003. https://doi.org/10.3390/ani11041003"

Answer:

Thank for your suggestion.  The two mentioned articles were cited and added to the references.

Reviewer comment:

"In the testis immunofluorescence results, the authors show higher immunopositivity for APEH in spermatozoa compared to Sertoli cells. They then showed a blot in which a higher quantity of this protease is evident in the Sertoli than in the spermatozoa. How do the authors explain this result?"

Answer:

Thank you for this observation.  The intensity of the fluorescence in immunofluorescence images are not indicative of quantity because this parameter can be influenced by several factors such as the laser settings, the resolution of the optic system to discriminate between two close immunofluorescence signals, the intrinsic characteristics of the preparation, among others.  For more details, please see https://www.olympus-lifescience.com/en/microscope-resource/primer/techniques/confocal/resolutionintro/.  Because of that, for quantification, only western blot bands should be considered.

Reviewer comment:

"In Figure 5A, the authors should put the bars in all figures"

Answer:

Done.  We have insert the new modified figure in the manuscript.

Round 2

Reviewer 3 Report

Accept as it is.